# Cognitive Functioning and Nail Salon Occupational Exposure among Vietnamese Immigrant Women in Northern California

**DOI:** 10.3390/ijerph19084634

**Published:** 2022-04-12

**Authors:** Thuc-Nhi Nguyen, Shuai Chen, Keith Chan, Mai Tram Nguyen, Ladson Hinton

**Affiliations:** 1Department of Psychology and Behavioral Sciences, University of California, Davis, CA 95616, USA; thucnhi4@gmail.com (T.-N.N.); shschen@ucdavis.edu (S.C.); iamngu@ucdavis.edu (M.T.N.); lwhinton@ucdavis.edu (L.H.); 2Silberman School of Social Work at Hunter College, CUNY, New York, NY 10035, USA

**Keywords:** Alzheimer’s disease factors, occupational hazards, exposure, cognitive functioning, Vietnamese Americans

## Abstract

**Introduction:** Vietnamese nail salon technicians are continuously exposed to neurotoxins linked to cognitive impairments and Alzheimer’s disease. This study examined the association of occupational exposure with cognitive function and depressive symptoms among Vietnamese nail salon technicians. **Methods:** The sample included 155 current or former Vietnamese female nail technicians and 145 control group participants. Measures included the Montreal Cognitive Assessment (MoCA) and the Center for Epidemiologic Studies Depression Scale (CES-D). **Results:** Average cognitive functioning was significantly higher for the control compared to the nail technician group (mean difference = 1.2, *p* < 0.05). No differences were observed for depression. Multivariate findings revealed that exposure was negatively associated with cognitive functioning (β = −0.29, 95% CI: −0.53, −0.05, *p* < 0.05). **Discussion:** Nail salon work and the extent of occupational exposure were associated with lower cognitive functioning among Vietnamese nail technicians. Longitudinal research can further examine the risk for cognitive decline and dementia for this vulnerable population.

## 1. Introduction

Occupational exposure to commonly used chemical substances in nail products, including organic solvents (e.g., toluene, formaldehyde, and dibutyl phthalate), plasticizers, resins, and acids, can pose significant risks for cognitive impairment [1]. Many of these products contain neurotoxic and carcinogenic properties known to cause cancer, reproductive problems, and central nervous system impairments that have been linked to Alzheimer’s disease [2]. Nail salon technicians are continuously exposed to them, and this exposure may occur at greater frequency and higher levels than in other occupational settings.

Despite growing evidence of the direct association between toxic chemical exposure and adverse health effects [3], the relationship between neurotoxin exposure and cognitive impairment for this population is still underexplored. Few studies have examined environmental risk factors at nail-care workplaces, and fewer focused on Vietnamese women nail technicians who overwhelmingly comprise the nail-care industry workforce. Vietnamese are the fourth largest Asian ethnic group in the US, following Chinese, Filipino, and Asian Indian [4]. Among these groups, they have the lowest socioeconomic status and highest risk for poor health and mental health [5]. In California, the cosmetology industry has the largest professional licensee population in the country. There are more than 300,000 technicians licensed to provide nail-care services, and approximately 80% are Vietnamese immigrant women [6]. Most technicians usually work 50 to 60 h per week and tend to remain on the job well into their senior years, where cognitive health problems brought on from their occupation can become more acute. A study in New York City found that Korean nail salon workers who were older, had worked longer in nail salons, and used less personal protective equipment, were significantly more likely to report deleterious health effects [7]. Significantly, the US Food and Drug Administration (FDA) does not require the cosmetic industry to test its products for safety. Of the several thousand chemicals and substances used in personal care products, about 90% have not been formally tested in the amounts and combinations used in the nail salon workplace [8].

The majority of research on neurotoxin exposure has been oriented towards industrial workplaces that routinely use hazardous chemicals and solvents: workplaces such as dental offices [9], electronic microscopy research labs [10], and among microelectronic technicians [11], house painters [12], and carpet layers [13]. Studies of these occupations found evidence of the association between neuropsychological/neurosensory deficits and neurotoxin exposure. Research on neuropsychology and cognition has further reinforced the acute health effects associated with low-level neurotoxin exposure, showing that they experienced more cognitive and psychological symptoms and had greater problems regarding attention deficits [14,15,16,17]. A growing body of research has indicated there are worrisome exposure levels to neurotoxic compounds in the nail-care industry. Kwapniewski et al., found a significantly high level of dibutyl phthalate in nail technicians’ urine samples [18], which may be twice as high as the general population [19]. Studies on neurotoxin exposure among nail workers found an increased risk of respiratory damage [20], skin irritation [21], and reproductive harm [22,23]. In another study, LoSasso et al., found evidence of neuropsychological and cognitive changes in a sample of 68 English-speaking non-immigrant women in the Detroit metropolitan area, indicating a potential risk of cognitive impairment [16]. Furthermore, a study that examined the exposure of nail salon workers in the Greater Boston area to semivolatile organic compounds (SVOCs), such as phthalates, phthalate alternatives, and organophosphate esters, found higher overall levels in post-shift urine concentrations for SVOC metabolites and on silicone wristbands worn during a work shift [24]. Overall, these studies indicate that continuous and long-term occupational exposure to neurotoxins can put nail salon workers at risk for physiological, neurological, and cognitive symptoms, which are consistent with existing literature in other occupations.

Despite the potential health and cognitive risks of nail salon work, relatively little is known regarding the occupational risks among Vietnamese women nail technicians who comprise a significant majority of the nail-care industry workforce. Past research has found that the incidence of dementia differs by race and ethnicity [25] and that socioeconomic status may account for the burden of risk factors among disadvantaged groups due to lower access to resources and appropriate medical care, along with a heightened risk for vascular disorders which contribute to differences in risk of dementia [26].

The current study adds to existing research by examining the association of nail salon occupational exposure with cognitive impairment and depressive symptoms in a vulnerable immigrant and refugee population. More specifically, we examined the association of neurotoxin exposure with cognitive impairment and depressive symptoms among two groups of Vietnamese women in Northern California: the nail technician group (NT group) and a demographically similar control group. We hypothesized that Vietnamese nail technicians would experience lower cognitive functioning and higher levels of depressive symptoms compared to a control group. This study addresses a critical gap in knowledge for a vulnerable population and provides further evidence for the link between neurotoxin exposure in nail salons and the potential risk of cognitive decline and Alzheimer’s disease.

## 2. Methods

### 2.1. Sample

The sample consisted of Vietnamese female participants aged 40 or above: a nail technician (NT) group (*n =* 155) and a control group (*n =* 145) who will be an approximate match with the NT group on demographic and acculturation variables.

### 2.2. Selection Criteria

NT group: Those who self-identify as a nail technician, hairstylist that works in a nail salon part-time, currently, or in the past, who is Vietnamese and female. Control group: Those who are Vietnamese, female, and have never worked in the cosmetic industry, including nail or hair salons.

### 2.3. Sampling and Recruitment

We used purposive and snowball sampling to select participants in the Northern California areas where there are high concentrations of Vietnamese and nail salons. We consulted with key community members to identify two community-based organizations that have a proven record with successful grant-funded research projects focused on nail salon sites: (blinded for review) and (blinded for review) in Northern California. Adopting practices from community-based research frameworks, we identified and received support from community liaisons through partnering organizations who are, or have been, nail technicians themselves or have family members working in the industry. Recruitment efforts included public announcements on popular Vietnamese-specific ethnic radio stations to help increase the study sample. In addition, the research team approached community-based organizations and coordinators for suggestions of possible participants, gathering a list of potential participants and their contact information. The research team contacted potential participants via phone stating the research’s purpose and obtained participants’ permission for further communication. The research team agreed to meet with participants at an appropriate location that was convenient and comfortable for participants to formally provide informed consent and share information relevant to the study. Recruitment at nail salons involved the research team first approaching the nail salon owners and explaining the objectives of this study to improve workers’ health. With their permission, the staff then approached workers for recruitment and data collection into the study. The final sample included 300 participants, with 155 who were current or former nail salon workers and 145 who did not work in nail salons. IRB approval was sought and obtained through the PI’s institution, and the review board approved the study after a careful review of the ethical considerations of the research.

### 2.4. Predictors

Occupational exposure: Self-reported workplace size (i.e., number of workstations), adequacy of ventilation (i.e., a self-rated scale, use of local ventilation), years of employment, and the number of hours of work per week. LoSasso et al. and Tabachinick et al. used similar proxies to estimate exposure to neurotoxicants and found a direct association between these variables and performance on neurobehavioral tests [16,17,27]. Demographics: Age, education, marriage status, and occupational status. Acculturation: English proficiency.

### 2.5. Outcome Measures

Cognitive functioning: The Montreal Cognitive Assessment (MoCA) is a commonly used 12-item cognitive screening instrument tool (scores range between 0 and 30 with higher scores indicating better cognitive functioning) that assesses short-term memory, visuospatial construction, executive functions, attention, concentration, language, and temporal and spatial orientation [28]. The instrument has a high test-retest reliability and good internal consistency in the general population and has been adapted and translated into Vietnamese [29]. Depression: The CES-D scale is a 20-item self-reporting scale that measures depressive symptoms and is used to screen depression and anxiety disorders [30]. The measure ranges from 0 to 60 and has been cross-culturally adapted into Vietnamese [31]. The scale has good reliability (α = 0.90). Both the MoCA and CES-D were chosen because they satisfied two important criteria for cultural equivalence in measurement, in capturing cognitive functioning and depression sufficiently well, specifically for the Vietnamese population, and for meaningful comparison with study samples from different cultural and ethnic groups in other studies [32].

### 2.6. Statistical Analyses

The primary outcome is the MoCA test score, and the secondary outcome is the CES-D depression scale. The primary interest lies in the association between the MoCA score and neurotoxin exposure. Characteristics of participants (demographic characteristics and outcomes) were summarized and compared between non-nail salon workers and nail salon workers. The Wilcoxon test was used to compare continuous and ordinal variables, and the χ^2^ test (or Fisher exact test) was used for categorical variables. Toxin exposure among nail salon workers was described by reporting the means (standard deviations) and counts (proportions). Multiple linear regression analysis was used to further evaluate the association between the outcomes and factors, where the primary interest was the association between the outcomes and exposure indices. The exposure index is defined as the product of years working in the current nail salon (or years working in the latest nail salon for former nail workers) and nail-related working hours per week. For non-nail workers, the exposure index is zero. Covariates included age, education, the status of living alone or not, and the ability to speak English. Diagnostic plots were examined, and no nonlinear patterns were noticed. Sensitivity analysis was also performed using the Huber–White robust standard error estimator in linear regression for potential heterogeneous variances. Based on the fitted regression, marginal predictive MoCAs (along with 95% CIs) were predicted at different values of the exposure index, averaging across other factors. Additional sensitivity analyses using linear regressions were performed in the subgroups of nail salon workers and current full-time nail salon workers, respectively.

There were 31 participants (10%) with missing values (most are due to missing years working in a nail salon or missing nail-related working hours, leading to a missing exposure index, especially among former nail salon workers), and these participants were excluded in the regression analysis. Sensitivity analysis was performed for linear regression based on multiple imputations for missing values using MICE (multiple imputations chained equations) under a missing-at-random assumption [33]. Data were analyzed using SAS 9.4 and Stata/SE 15. A two-sided *p* < 0.05 was used to determine statistical significance.

## 3. Results

### 3.1. Participant Characteristics

Of the 300 participants, there were 155 nail salon workers and 145 participants who had not worked as nail salon workers. Table 1 summarized the characteristics of participants and compared them between non-nail salon and nail salon workers. All participants were female Vietnamese, and the average age was 58.0 (SD ± 6.2) years. The average MoCA score was 18.8 (SD ± 4.5) for non-nail salon workers, which is significantly higher than 17.6 (SD ± 4.3) for nail salon workers (*p* = 0.017). Similarly, the education-adjusted MoCA score was significantly higher for non-nail salon workers (control group) and nail salon workers (*p* = 0.021). However, there is no significant difference between the two groups on the average CES-D (depression scale) (*p* = 0.22). Appendix A further summarized the characteristics of current full-time, former, and part-time nail salon workers, respectively. Among the 155 nail salon worker participants, there were 136 current full-time, 5 part-time, and 14 former nail salon workers. Given the fact that the former nail salon workers were less than 10% of all nail salon workers, the potential recall bias from former nail salon workers was expected to be mild.

Of the 155 nail salon workers who participated in the study, Table 2 summarized their toxin exposure. The average years working in the current nail salon was 6.2 (SD ± 6.6) years. The proportions of self-rated quality of environment were 48.3% as good/great, 40.9% as normal, and 10.7% as horrible/bad. The proportions of always wearing masks and gloves were 33.6% and 68.9%, respectively. Appendix A further summarized the toxin exposure of current full-time, former, and part-time nail salon workers, respectively.

### 3.2. Association between MoCA Score and Factors

Table 3 summarizes the association between the MoCA score and factors. Results are presented as the adjusted estimates of coefficients for factors based on multiple linear regression. A higher exposure index (in 100) was significantly associated with a lower MoCA score, with the estimated coefficient as −0.29 (95% CI = (−0.53, −0.05), *p* = 0.017). That is, with an increase of 100 in the exposure index (years working in current nail salon × hours worked per week), the MoCA score is expected to decrease by 0.29. Higher education was also significantly associated with a higher MoCA score. Living alone, the ability to speak English and age were not significantly associated with MoCA scores after an adjustment by other demographic and acculturation factors. Figure 1 displayed the marginal predictive MoCAs at different values of exposure index, which also demonstrated a decreasing MoCA score with an increasing exposure index. Appendix A reported sensitivity analysis results based on the Huber–White robust standard error estimator and multiple imputations, respectively, with significance remaining. Sensitivity analysis in the subgroups (Appendix A) showed that the MoCA score was expected to decrease by 0.19 with an increase of 100 in the exposure index among nail salon workers only, and decrease by 0.22 among current full-time nail salon workers only, although both are no longer significant in subgroups.

## 4. Discussion

To our knowledge, this study is the first to examine nail salon exposure as a risk factor for poor cognitive functioning and depression among Vietnamese. Findings from this study indicated that working as a nail salon technician was associated with lower cognitive functioning among middle-aged Vietnamese women, as measured by the MoCA. In addition, the extent of exposure to environmental hazards, as measured by the number of years of work in a nail salon and the hours worked per week, was associated with lower cognitive functioning, even when controlling for demographic and acculturation factors. While our study did not directly measure neurotoxin exposure, our findings are consistent with past research indicating the risk of cognitive impairment from long-term exposure to neurotoxins in occupations, such as dental technicians and carpet layers [1,2,3,9,13].

The lack of association between depressive symptoms and nail salon workplace exposure is discrepant with a number of prior studies [10,11,12]. Although our study’s findings did not indicate a significant association between nail salon workplace exposures to depressive symptoms, past research has highlighted the relationship between cognitive impairment and poor mental health. One possible explanation is that our study only assessed depressive symptoms while other studies have examined a broader range of neuropsychiatric symptoms. It is also possible that a history of nail salon employment is a proxy for better socioeconomic circumstances and is a protective element for depression. Although our findings suggested that nail salon work was associated with a risk for cognitive impairment, the women who engaged in this occupation may have attained financial stability and flexibility as immigrant entrepreneurs, which may have been protective against depression in later life. This is consistent with past research which found that socioeconomic status is protective against cognitive decline and dementia for older adults [26].

Although this study has many important strengths, there are several limitations. The study used purposive, snowball sampling specifically in Northern California, and the generalizability of the study results to the larger population of Vietnamese women nail technicians in the US may need further consideration. Another limitation is that the analyses did not account for the potential clustering effects (e.g., nail salon, family, neighbor community, etc.) due to unavailable information about cluster identification within neighborhoods. In addition, our study design is cross-sectional in nature, which limits conclusive determinations in the directionality of occupational exposure and cognitive functioning. More in-depth analysis regarding the association of nail salon work and loss in specific dimensions of cognition functioning, such as memory, attention, and executive functioning, can provide further insights. However, our findings are consistent with past research which has suggested that the frequency and degree of exposure to neurotoxins, such as phthalates and other semivolatile organic compounds, may contribute to greater risk [24]. Although we were unable to directly measure exposure, our findings meaningfully contribute to the evidence that any exposure through working in nail salons may present risk factors for cognitive decline, even when controlling for socioeconomic and acculturation variables.

Despite these limitations, our study provided important and novel insights into the relationship between occupational exposure, nail salon work, and cognitive functioning among a vulnerable immigrant refugee population. Future longitudinal research can examine exposure to specific neurotoxins in the workplace and the risk of cognitive decline, which would confirm and extend our findings. This study highlights the need for future consideration of community-based, targeted strategies to prevent occupational exposure to toxins for Vietnamese women employed in the nail-care workplace environment. Geriatricians and other providers who work with this population can conduct more comprehensive screening for cognitive changes that may be related to workplace environmental hazards. Public policy can advocate for better protective equipment and ventilation in these types of workplace settings. Public service announcements that are targeted to Vietnamese communities can increase awareness of the long-term effects of neurotoxins associated with nail salon work.

## 5. Conclusions

This study provided findings on the relationship between nail salon work, cognitive functioning, and symptoms of depression among a sample of community-dwelling Vietnamese women living in Northern California. Our findings are consistent with past research indicating the risk of cognitive impairment in occupations from long-term exposure to neurotoxins and provide insight into the risks from exposure for Vietnamese women who are over-represented in the nail salon industry. Nail salon work has provided a source of income for many Vietnamese who identify as refugees, as a way to support one’s family that provided flexibility and had relatively low start-up costs, even for those with little to no formal education. As involuntary migrants who escaped persecution, nail salon work is a means of survival that requires no prior training and can be performed with little English ability. Our findings indicated that Vietnamese women who worked in the nail salon industry may experience risks in their cognitive functioning, although these results should be taken with caution because we did not control for factors that were not included in this study, such as levels of neurotoxins and possible exposure in other settings. Further research is needed to investigate the relationship between acculturation and exposure to neurotoxins through data collection on biomarkers, cognitive functioning, and mental health for this understudied and underserved population.

## Figures and Tables

**Figure 1 ijerph-19-04634-f001:**
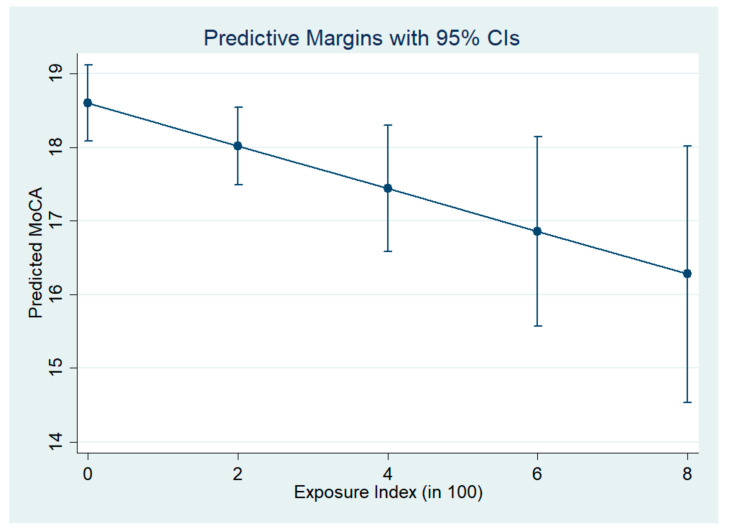
Predictive margins for the Montreal measure of cognitive assessment (MoCA) and 95% CIs at different values of exposure index averaged across other factors, based on the fitted multiple linear regression (*n* = 269). Exposure index = (years working in current nail salon) × (hours worked per week). For non-nail workers, exposure index = 0.

**Table 1 ijerph-19-04634-t001:** Characteristics of participants.

	Non-Nail Salon Worker (*n =* 145)	Nail Salon Worker ^a^ (*n =* 155)	Total (*n =* 300)	*p*-Value ^b^
	**N(%)**	**N(%)**	**N(%)**	
**Education**				0.585
Elementary/middle school	49 (34.0%)	44 (29.1%)	93 (31.5%)	
High school	42 (29.2%)	61 (40.4%)	103 (34.9%)	
Technical school/Some college	30 (20.9%)	36 (23.8%)	66 (22.4%)	
Bachelor/Master/Ph.D/MD/JD etc.	23 (16.0%)	10 (6.6%)	33 (11.2%)	
**Marriage status**				0.128
Single	18 (12.5%)	10 (6.6%)	28 (9.5%)	
Married	82 (56.9%)	90 (59.6%)	172 (58.3%)	
Divorced	16 (11.1%)	22 (14.6%)	38 (12.9%)	
Widow	23 (16.0%)	17 (11.3%)	40 (13.6%)	
Domestic partner	5 (3.5%)	12 (8.0%)	17 (5.8%)	
**Ability to speak English**				0.111
Little	21 (14.5%)	9 (6.0%)	30 (10.1%)	
Average	75 (51.7%)	84 (55.6%)	159 (53.7%)	
Good/Great	49 (33.8%)	58 (38.4%)	107 (36.2%)	
	**Mean (±SD)**	**Mean (±SD)**	**Mean (±SD)**	
**Age**	58.7 (±6.5)	57.4 (±6.0)	58.0 (±6.2)	0.126
**MoCA score**	18.8 (±4.5)	17.6 (±4.3)	18.2 (±4.4)	**0.017**
**Education-adjusted MoCA score ^c^**	19.1 (±4.4)	18.0 (±4.2)	18.5 (±4.3)	**0.021**
**CES-D (Depression scale)**	15.7 (±11.1)	16.6 (±9.8)	16.2 (±10.4)	0.218

^a^ Nail salon workers include 136 current full-time, 14 former, and 5 part-time nail salon workers. ^b^
*p*-values of comparing non-nail and nail workers with Wilcoxon rank sum tests for continuous and ordinal variables, and χ^2^ test (or Fisher exact test) for categorical variables. ^c^ If a participant did not finish high school, the education-adjusted MoCA score = MoCA + 1; otherwise, education-adjusted MoCA score = MoCA.

**Table 2 ijerph-19-04634-t002:** Summary of toxin exposure among all nail salon workers (*n =* 155). Raw means (standard deviations) and counts (proportions) are reported.

	Observed n	Mean (±SD) (Range) or N (%)
**Years working in current nail salon**	133	6.2 (±6.6) (0.04–27)
**Hours worked per week**	144	32.9 (±15.1) (1–70)
**Exposure Index ^a^**	125	206.3 (±260.1) (1.25–1500)
**Environment in current workplace**		
Manitable	140	5.0 (±2.4) (0–12)
Pedichair	145	5.7 (±3.2) (0–15)
Lunch area	145	126 (86.9%)
Eating at the right time every day	121	49 (40.5%)
Front door	143	130 (90.9%)
Back door	142	52 (36.6%)
Table vent	130	80 (61.5%)
Table fan	139	125 (89.9%)
Ceiling fan	141	40 (28.4%)
Air condition	137	121 (88.3%)
Stationary vent	125	67 (53.6%)
Filtration	129	51 (39.5%)
**Self-rated quality of environment**		
Horrible	149	3 (2.0%)
Bad	149	13 (8.7%)
Normal	149	61 (40.9%)
Good	149	66 (44.3%)
Great	149	6 (4.0%)
**Work position**		
Staff	149	127 (85.2%)
Owner	149	22 (14.8%)
**Service that current workplace offered**		
Pedi	146	146 (100.0%)
Mani	145	145 (100.0%)
Silk	133	62 (46.6%)
Gel	148	141 (95.3%)
Acrylics	147	135 (91.8%)
Extra	148	111 (75.0%)
**Service that participants offered**		
Pedi	149	144 (96.6%)
Mani	148	142 (96.0%)
Silk	139	32 (23.0%)
Gel	147	104 (70.8%)
Acrylics	144	68 (47.2%)
Extra	132	51 (38.6%)
**Mask**		
Never	149	33 (22.2%)
Rarely	149	17 (11.4%)
Often	149	49 (32.9%)
Always	149	50 (33.6%)
**Gloves**		
Never	148	9 (6.1%)
Rarely	148	4 (2.7%)
Often	148	33 (22.3%)
Always	148	102 (68.9%)

^a^ Exposure index = (years working in current nail salon) × (hours worked per week).

**Table 3 ijerph-19-04634-t003:** Association between MoCA and factors using multiple linear regression in all participants without missing values in variables (*n =* 269).

	Estimate (95% CI)	*p*-Value
**Education**		
Elementary/middle school	Reference	
High school	2.92 (1.73, 4.11)	**<0.001**
Technical school/Some college	4.61 (3.19, 6.03)	**<0.001**
Bachelor/Master/Ph.D/MD/JD etc.	6.24 (4.52, 7.97)	**<0.001**
**Live alone**		
Yes	Reference	
No	0.02 (−0.98, 1.01)	0.975
**Ability to speak**		
Little	Reference	
Average	1.35 (−0.24, 2.95)	0.098
Good/Great	1.35 (−0.44, 3.15)	0.141
**Age (in years)**	−0.07 (−0.14, 0.01)	0.084
**Exposure index (in 100) ^a^**	−0.29 (−0.53, −0.05)	**0.017**

^a^ Exposure index = (years working in current nail salon) × (hours worked per week). For non-nail workers, exposure index = 0.

## Data Availability

Data was collected and archived by the PI as required by funding requirements of the Alzheimer’s Association.

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
