# Peer review of "Cognitive Functioning and Nail Salon Occupational Exposure among Vietnamese Immigrant Women in Northern California"

_ijerph, 2022, doi:10.3390/ijerph19084634_

Round 1
Reviewer 1 Report
The manuscript discusses the effects of occupational chemical exposure on the cognitive function of 155 Vietnamese immigrants working in nail salons. The cognitive function is measured through MoCA and CES-D methods. The aim of the research is clear and sound. The manuscript is well written and well organized but needs an improvement by adding a conclusion. However, from the discussion, we failed to understand the relation between the related chemical exposure to the quality of cognitive function. Despite the authors' claim that their results provide important and novel insights on the relationship between occupational exposure, nail salon work, and cognitive function, it is also pointed out that the study finds a less significant association between nail salon workplace exposure to the depressive symptoms. It is claimed that the results suggest the relationship between nail salon work with risk for cognitive impairment, but we found from Table 1 that all MoCA, Education-adjusted MoCA, and CES-D scores between Non-Nail Salon (as the control group) and Nail Salon Worker show no significant differences. Please explain this conclusion clearly. Our next concern is related to the equality between the Nail Salon Worker and the control group. It is shown that education levels between both groups are not comparable. For instance, the Bachelor/Master/Ph.D/MD/JD level is 16% for the control group, while only 6% for the Nail Salon Worker. This difference may lead to an unobjective conclusion regarding cognitive function. The authors should probably consider adding new participants for the control group to make the related education level relatively comparable. After these major revisions, we recommend the publication of the manuscript.
Author Response
Thank you for the comments on the research aim and writing of this manuscript. We were careful to present a study that was meticulous, well written and well organized. We decided to add a conclusion section based on the reviewer’s suggestion, specifically regarding the association of nail salon work and exposure to chemicals with cognitive functioning.
In terms to results from Table 1, the mean MoCA score was reported to be significantly different for non-nail salon workers (control group) and nail salon workers at p = 0.017. Similarly, the education-adjusted MoCA score was significantly different for non-nail salon workers (control group) and nail salon workers at p = 0.021. We have clarified this by adding another sentence in the results section based on the reviewer’s feedback.
The question on equality between the nail salon worker and control group is an interesting one, where non-nail salon workers reported overall higher education. This can be a factor in cognitive functioning, though we accounted this through the use of an education-adjust MoCA score and found a statistically significant difference between the two groups. There is some historical and cultural context that may be important to highlight, in that Vietnamese who first arrived as refugees in the US became nail salon workers because it was a way to support one’s family that provided flexibility and had relatively low start-up costs even for those with little to no formal education. As involuntary migrants who were escaping persecution, nail salon work was a means of survival that required no prior training and can be performed with little English ability. It is likely that there was some selection bias for nail salon workers who found the profession to be one that supported their need to provide financially for their families while allowing them to continue to parent their children. Further research can be done to investigate this as a function of Vietnamese families in the context of acculturation, cognitive functioning and mental health.
Reviewer 2 Report
This is an interesting study about the association of occupational exposure with cognitive function and depressive symptoms among Vietnamese nail salon technicians. Authors tested Montreal Cognitive Assessment 12 (MoCA) and the Center for Epidemiologic Studies Depression Scale (CES-D) in 155 current or former Vietnamese female nail technicians and 145 control group participants. Authors found that average cognitive functioning (as shown by MoCA) was significantly higher for control compared to nail technician group, while no difference was observed for depression. The Authors conclude that nail salon work and extent of occupational exposure might be associated with lower cognitive functioning among their Vietnamese sample.
Overall, the topic investigated is innovative and the study is well-designed. This contribution is surely suited for the publication in International Journal of Environmental Research and Public Health. Few comments should however be addressed:
- I suggest Authors including some graphs showing the effects they found to allow a better understanding of the results.
- It would be also interesting to investigate whether neurotoxin might be considered a factor mediating the effect of age and education on MoCA score. Maybe the data could be analyzed to investigate this issue, otherwise they could report this point as a possible future direction.
- Moreover, it would be interesting to investigate in detail what kind of cognitive function was damaged in this sample: memory, attention, executive functions? Discuss this aspect as a limit of the study.
- Since Authors did not measure the level of neurotoxin in the sample, they should underline in the Discussion that conclusions should be taken with caution, because we can not rule out that other factors not controlled in this research could impact on the results (e.g., the groups could be different for a number of characteristics and not only for the presence or absence of neurotoxin). Moreover, in the Discussion they should underscore that we can not conclude that a causal relationship could be present between neurotoxin level and MoCA score, since the experimental design does not allow to draw such a conclusion.
After having addressed these points, I think this study could deserve the publication in International Journal of Environmental Research and Public Health.
Author Response
Overall, the topic investigated is innovative and the study is well-designed. This contribution is surely suited for the publication in International Journal of Environmental Research and Public Health. Few comments should however be addressed:
- I suggest Authors including some graphs showing the effects they found to allow a better understanding of the results.
Based on the suggestion of the reviewer we have included Figure 1 which highlights the relationship of the exposure index to predicted MoCA.
- It would be also interesting to investigate whether neurotoxin might be considered a factor mediating the effect of age and education on MoCA score. Maybe the data could be analyzed to investigate this issue, otherwise they could report this point as a possible future direction.
In regard to neurotoxins as a factor in the effect of age and education, this would be a very interesting investigation given that there is likely a longitudinal interaction of exposure with age and education on cognitive decline. However, this is outside the scope of the current study, and due to the cross-sectional data collection, this type of analysis would not be possible with the data. We hope this can be a project we can pursue through future funding, although longitudinal data collection (which we hope we can collect along with biomarkers) would be very expensive but we believe would be worthwhile.
- Moreover, it would be interesting to investigate in detail what kind of cognitive function was damaged in this sample: memory, attention, executive functions? Discuss this aspect as a limit of the study.
On what kind of cognitive function was damaged in this sample: memory, attention, executive functions, we described the MoCA on page 7 as measuring “assesses short-term memory, visuospatial construction, executive functions, attention, and concentration, language, and temporal and spatial orientation.” This instrument was psychometrically validated and adapted and translated into Vietnamese. Confirmatory factor analyses were conducted on the construct validity of items as they relate to dimensions of memory, attention and executive functions, as we hope to publish these results as another manuscript in the future. We have included a sentence on page 12 to describe this as a limitation and a future direction for more in-depth analysis.
- Since Authors did not measure the level of neurotoxin in the sample, they should underline in the Discussion that conclusions should be taken with caution, because we can not rule out that other factors not controlled in this research could impact on the results (e.g., the groups could be different for a number of characteristics and not only for the presence or absence of neurotoxin). Moreover, in the Discussion they should underscore that we can not conclude that a causal relationship could be present between neurotoxin level and MoCA score, since the experimental design does not allow to draw such a conclusion.
We have included a new conclusion section on page 13, where we also highlighted caution in the interpretation of results because we did not control for factors that were not included in this study such as level of neurotoxin and possible exposure in other settings. We conclude by saying that further research is needed to investigate the relationship of acculturation, exposure of neurotoxin through data collection on biomarkers, cognitive functioning and mental health for this understudied and underserved population.
After having addressed these points, I think this study could deserve the publication in International Journal of Environmental Research and Public Health.
Thank you for the opportunity for consideration in this publication.
Reviewer 3 Report
The article "Cognitive Functioning and Nail Salon Occupational Exposure among Vietnamese Immigrant Women in Northern California" reports neurotoxic effects of nail salon products (phtalate volatile organics) on salon personnel due to prolonged exposure. The study utilizes MoCA and CESD methods to estimate cognitive function and depression/anxiety disorders among the subjects. The work is performed thoroughly and the data support the claims. I have the following comments/concerns that could be considered to improve the manuscript.
1. An ethical statement and approval of protocol needs to be stated in the methods section.
2. Is it possible to mention the specific neurotoxicants associated with manicure/pedicure/acrylics in Table 2? Duration of exposure (work hours) would be particularly useful in this aspect.
3. Is there any available data on any pre-existing health conditions/trauma/accident records in the NT population that possible would interfere with psychiatric symptoms? Especially, if the subjects were taking anti-depressants/nerve stimulants at the time of study might be a screening factor.
4. For MoCA tests, were there any pre-conditioning performed with the subjects before the actual tests were run? Can the language of the test be a factor for subjects who are not fluent in English?
5. General remark about CES-D study for depression and anxiety - was the cutoff range set based on IQ scores? Also, is it possible the economic conditions of subjects might need to be factored into an a priori comparison? Besides, sample size might be too small for a CES-D outcome.
Thank you for the review opportunity.
Author Response
- An ethical statement and approval of protocol needs to be stated in the methods section.
We included a statement on page 7 in the methods section as suggested.
- Is it possible to mention the specific neurotoxicants associated with manicure/pedicure/acrylics in Table 2? Duration of exposure (work hours) would be particularly useful in this aspect.
We reported on the specific neurotoxins that are commonly found in nail salon work on page 4. We hesitate to report on them specifically in Table 2 because we did not conduct lab research on the specific chemicals that were reported in the products that were used. This is a very interesting observation from the reviewer, and lab-based research to examine the specific type of chemical in various products can lend important insights for policy and prevention. This is however outside the scope of our current study.
- Is there any available data on any pre-existing health conditions/trauma/accident records in the NT population that possible would interfere with psychiatric symptoms? Especially, if the subjects were taking anti-depressants/nerve stimulants at the time of study might be a screening factor.
The sample is community-dwelling and we did not obtain IRB approval for their specific medical records, though having data on pre-existing health conditions/trauma/accident records would lend greater insights to the causal pathways of cognitive functioning and depression for this population. Although we did not have this pre-existing data, sensitivity analyses using linear regressions were performed in the subgroups of nail salon workers and current full-time nail salon workers, respectively, to control for possible effects. Our analyses found that the control and NT population were similar in demographic characteristics, though having pre-existing data would be preferable as the reviewer pointed out.
- For MoCA tests, were there any pre-conditioning performed with the subjects before the actual tests were run? Can the language of the test be a factor for subjects who are not fluent in English?
As described on page 7, the MoCA was validated, adapted and translated into Vietnamese through a rigorous cross-cultural validation process.
- General remark about CES-D study for depression and anxiety - was the cutoff range set based on IQ scores? Also, is it possible the economic conditions of subjects might need to be factored into an a priori comparison? Besides, sample size might be too small for a CES-D outcome.
Thank you for the review opportunity.
The cutoff scores in the CES-D were set by the creators of the scale, based on its ability to detect persons at risk for major depression and validated through testing. In terms of economic conditions, we controlled for education and occupational status as well as English proficiency.
Thank you for considering this manuscript for publication.
Round 2
Reviewer 1 Report
We agree with the revision made by the authors and we can recommend this manuscript for publication.